

# Clinical outcomes of residual or recurrent nasopharyngeal carcinoma treated with endoscopic nasopharyngectomy plus chemoradiotherapy or with chemoradiotherapy alone: a retrospective study

Jingjin Weng[1,2], Jiazhang Wei[2], Jinyuan Si[2], Yangda Qin[2], Min Li[2], Fei Liu[3], Yongfeng Si[2] and Jiping Su[1]

[1] Department of Otolaryngology-Head and Neck Surgery, First Affiliated Hospital of Guangxi Medical University, Nanning, China
[2] Department of Otolaryngology & Head and Neck Oncology, The People's Hospital of Guangxi Zhuang Autonomous Region, Nanning, China
[3] Research Center of Medical Sciences, The People's Hospital of Guangxi Zhuang Autonomous Region, Nanning, China

Corresponding authors
Yongfeng Si, syfklxf@126.com
Jiping Su, ymsu2@126.com

## ABSTRACT

**Background.** Local residual and recurrent nasopharyngeal carcinoma (NPC) generally shows treatment failure after standard radiotherapy with or without concurrent chemotherapy. Whether endoscopic nasopharyngectomy might provide an additional therapeutic advantage remains controversial. Therefore, we retrospectively compared the clinical prognoses of patients with residual or recurrent NPC treated with endoscopic nasopharyngectomy combined with chemoradiotherapy (CRT) with those of patients treated with CRT alone.

**Methods and Materials.** A total of sixty-two patients with local residual or recurrent NPC were studied retrospectively: 36 patients received endoscopic nasopharyngectomy combined with CRT, whereas 26 patients who refused the surgery or had surgical contraindications received CRT alone. Serum Epstein-Barr virus (EBV) DNA levels were measured pre- and post-treatment. The differences in prognosis between the two treatment regimens and the pre- and post-treatment changes in EBV-DNA levels were analyzed.

**Results.** The median follow-up time was 31 months, with a 3-year overall survival (OS) of 51.40% and a 3-year disease-free survival (DFS) of 46.86%. The surgery + CRT group had a better OS than the CRT alone group did ($\chi^2 = 4.054$, $P = 0.044$). The pretreatment EBV-DNA levels showed a positive correlation with the clinical staging of recurrent NPC ($\chi^2 = 11.674$, $P = 0.009$). Patients with negative pretreatment serum EBV-DNA levels showed a superior OS to those of patients who tested positive for EBV-DNA ($>0$ copy/mL) ($\chi^2 = 9.833$, $P = 0.002$). The post-treatment EBV-DNA levels, compared with the pretreatment levels, decreased significantly in the surgery + CRT group ($Z = -3.484$, $P = 0.000$). In contrast, the EBV-DNA levels after CRT alone did not decrease significantly ($Z = -1.956$, $P = 0.051$). Multivariate analysis indicated that local staging, pretreatment EBV-DNA load, and the treatment method

were independent risk factors for OS. Subgroup analysis indicated that the patients who tested negative for EBV-DNA before the treatment and those who received surgery + CRT showed a better OS than those who received CRT alone.

**Conclusions.** The pretreatment serum EBV-DNA level was associated with disease prognosis. The combination therapy preceded by surgery can effectively decrease the copy number of EBV-DNA. Patients with local intermediate- and late-stage NPC, especially those negative for EBV-DNA, may consider opting for surgery followed by post-operative adjuvant radiotherapy or chemotherapy.

# INTRODUCTION

Nasopharyngeal carcinoma (NPC) is a common malignant disease in the southern regions of China. Because NPC is sensitive to irradiation during the initial treatment, radiotherapy is the preferred treatment. However, approximately 10% of patients show local recurrence after radiotherapy (*Yu et al., 2005*). After the first course of radiotherapy, patients often present with fibrous hyperplasia of the nasopharynx and poor local circulation due to local vascular occlusion. This condition can lead to low efficacy of the treatment and many side effects from re-irradiation alone. In addition, the residual tumor after the first treatment may acquire mutations or some other mechanisms that are likely to persist and render the tumor resistant to the same treatment. Therefore, surgical treatment of residual and recurrent NPC has been advocated (*Chan & Wei, 2012*).

Compared with the conventional maxillary swing approach for the resection of recurrent tumors, endoscopic nasopharyngectomy is less invasive and does not result in facial scars. Therefore, a number of institutions have begun performing endoscopic nasopharyngectomy for the resection of residual and recurrent NPC (*Chen et al., 2009*; *Zou et al., 2015*). However, a previous study has shown that the serum level of Epstein-Barr virus DNA (EBV-DNA) is an effective indicator for monitoring the therapeutic efficacy in patients with recurrent NPC (*An et al., 2011*). Nonetheless, there is still a group of patients with residual or recurrent NPC in whom EBV-DNA cannot be detected. The suitability of surgery for these patients and whether surgery can be performed in late-stage recurrent NPC have rarely been reported in the literature. Thus, this retrospective study was carried out to determine the plasma levels of EBV-DNA in patients with residual or recurrent NPC to explore any variations and their association with the disease prognosis.

# MATERIALS AND METHODS

## Study criteria and patient characteristics

Data for the patients with residual or recurrent NPC were retrieved from the database of The People's Hospital of Guangxi Zhuang Autonomous Region (Nanning, China). All the patients recruited in this study provided written informed authorization consent, in which

they agreed to use of their clinical and imaging data for non-commercial scientific research. We adhered to the bioethics principles of the Declaration of Helsinki. This retrospective study was approved by the ethics committee of The People's Hospital of Guangxi Zhuang Autonomous Region (Ethical Application Ref: Keyan-Guangxi-Keji-2016-31).

The case inclusion criteria included (1) patients admitted to The People's Hospital of Guangxi Zhuang Autonomous Region with residual or recurrent NPC. The diagnoses of residual or recurrent NPC for every case included in our study were strictly confirmed by pathological examination. The staging for each patient was based upon the T classification of residual or recurrent tumors. A residual tumor was defined as a persistent lesion in the nasopharynx within six months after full-dose radiotherapy. A recurrent tumor was defined as a local recurrence after complete remission within six months after full-dose radiotherapy and the subsequent emergence of new lesions in the nasopharynx after six months. Additional inclusion criteria were (2) a measurable lesion with evaluable efficacy, (3) a Karnofsky score ≥70, and (4) provision of written informed consent to undergo surgery and/or chemoradiotherapy. The case exclusion criteria were (1) patients with cervical lymph node metastasis or distant metastasis, (2) patients with severe damage to liver and kidney function, and (3) patients with severe immune deficiency.

A total of 62 patients with residual or recurrent NPC who were treated at our hospital between June 2011 and April 2013 were enrolled in the study (five cases of residual tumors and 57 cases of recurrent tumors). The patients included 47 men and 15 women, aged 22 to 70 years with a median age of 50 years. Of these 62 patients, 58 patients were pathologically identified as having undifferentiated non-keratinizing carcinoma (WHO type III), three patients had keratinizing squamous cell carcinoma (WHO type I), and only one patient had differentiated non-keratinizing carcinoma (WHO type II).

In general, surgery was recommended and preferred for all the regional residual or recurrent cases, except when the following exclusion criteria were encountered: (1) internal carotid artery encasement, massive intracranial intradural involvement, or orbital space invasion; (2) uncontrolled nasopharyngeal skull infection; (3) general anesthesia surgery contraindications; or (4) patient refusal to undergo surgery. To retrospectively analyze the clinical outcome of the patients who underwent endoscopic nasopharyngectomy, the patients were divided into two groups according to the treatment selection. In the surgical group, 36 patients underwent complete endoscopic nasopharyngectomy of the residual or recurrent tumor and subsequent post-operative chemoradiotherapy (CRT). In the CRT alone group, 26 patients received concurrent radiotherapy and chemotherapy. The patients' sex, age, local staging, and EBV-DNA positive rate showed no statistically significant differences between the two groups (all $P > 0.05$) (Table 1).

## Treatment methods

The surgical group received endoscopic nasopharyngectomy plus concurrent radiotherapy and a chemotherapy treatment regimen. Surgery was performed under general anesthesia. The patients were placed in a supine position, and a cotton swab soaked with epinephrine was used to shrink the nasal mucosa 2–3 times. Under the guidance of nasal endoscopy, an electric knife was then used to resect the nasopharyngeal tumor via a nasal approach.

**Table 1  Comparison of clinical data between the surgical group and CRT group.**

| Factors | Surgical group | CRT group | $\chi^2$ | *P* value |
|---|---|---|---|---|
| Sex | | | | |
|     Male | 26 | 21 | | |
|     Female | 10 | 5 | 0.601 | 0.438 |
| Age | | | | |
|     <50 years | 19 | 12 | | |
|     ≥50 years | 17 | 14 | 0.265 | 0.607 |
| Local staging | | | | |
|     rT1 | 8 | 2 | | |
|     rT2 | 9 | 10 | | |
|     rT3 | 8 | 9 | 4.465 | 0.215 |
|     rT4 | 11 | 5 | | |
| EBV-DNA group | | | | |
|     Negative | 19 | 16 | | |
|     Positive | 17 | 10 | 0.471 | 0.492 |

**Notes.**
Abbreviations: CRT, chemoradiotherapy.

The resection margin was defined according to the results of nasopharyngeal skull-base magnetic resonance imaging (MRI) and intraoperative observations. Typical imaging data of the patients are shown in Fig. 1. According to post-operative pathological examinations, all the resections were confirmed to be residual or recurrent tumors and three patients in the surgery group (8.3%, 3/36) were found to have positive margins.

The chemotherapy regimen and doses for the surgical + CRT and CRT alone group were the same and consisted of 30 mg/m$^2$ cisplatin via intravenous infusion beginning on days 1–3 and 1,000 mg/m$^2$ gemcitabine via intravenous infusion on day 1 and day 8; each cycle lasted 21 days, and 2–4 cycles were administered consecutively. For the intensity-modulated radiotherapy (IMRT) for both groups, the target area was delineated on the basis of the tumor boundary shown in MRI, where in the gross tumor volume (GTV) was irradiated with 2.0 Gy/fraction 5days/week; the total dose of GTV was 60–66 Gy. The median radiotherapy dose for primary NPC was 71.2 (52–81) Gy. The median time span between the primary treatment and treatment of the recurrence was 20 (3–178) months.

## Collection of peripheral blood samples and EBV-DNA load assay

A 2 mL blood sample was collected from each patient's antecubital vein pre- and post-treatment in an anticoagulation tube containing EDTA, centrifuged to separate the blood plasma into the upper phase for DNA extraction, and stored at −20 °C for later use. A Light-Cycler 480 quantitative fluorescence PCR platform (Roche Diagnostics, Basel, Switzerland) was used to quantify EBV-DNA levels. The test kit was provided by the Da'an Gene Diagnostic Center of Sun Yat-Sen University (Cat. # DA-D065). The target gene for amplification was derived from the *BamHI*-W fragment of EBV. All procedures were performed according to the manufacturer's protocol. EBV-DNA >0 copies/mL was

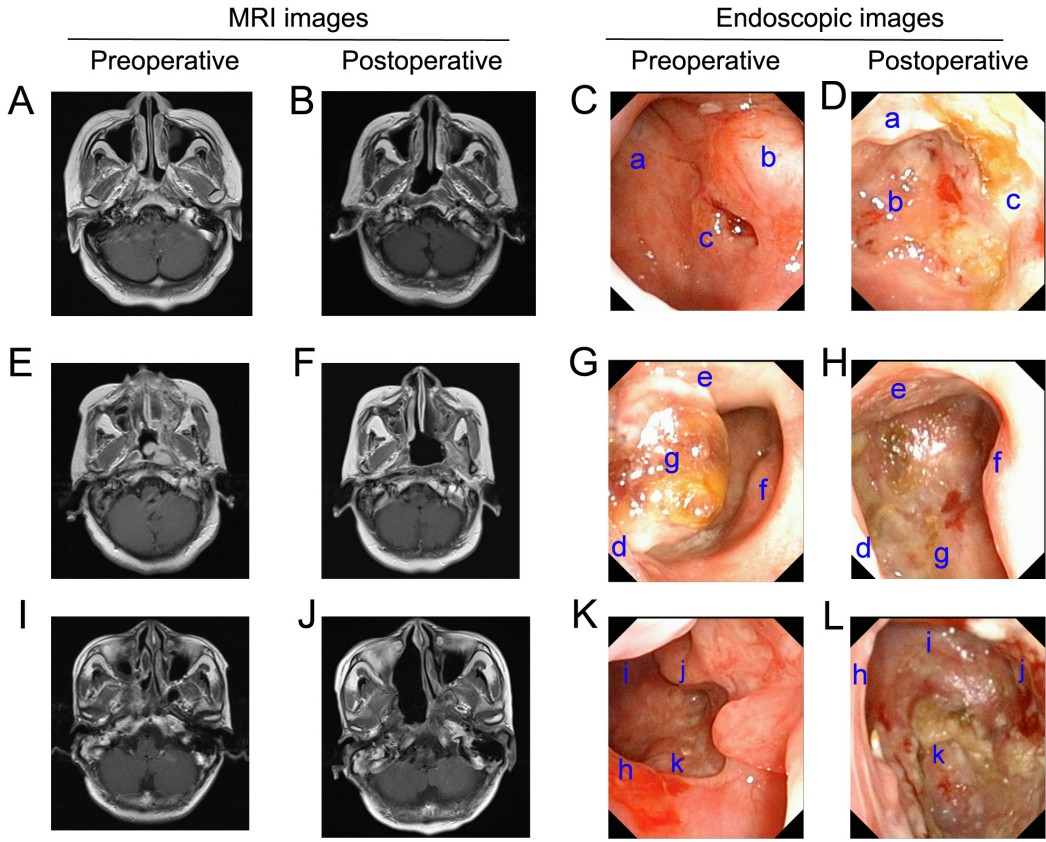

**Figure 1  Pre- and post-operative MRI and high-definition endoscopic images.** (A) Pre-operative MRI shows that the tumor is located in the right pharyngeal recess (rT1). (B) The 6-months post-operative MRI did not show tumor recurrence. (C and D) The 6-months post-operative endoscopic examination images show no visible tumor recurrence. Labels in the pre-operative endoscopic images: a, nasopharyngeal posterior wall; b, right torus tubarius; c, nasopharyngeal carcinoma. Post-operative: a, soft palate; b, nasopharyngeal posterior wall; c, nasopharyngeal right wall. (E) Pre-operative MRI shows the tumor invading the left pharyngeal space (rT2). (F) Post-operative MRI after six months did not show tumor recurrence; (G & H) The 6-months post-operative endoscopic examination images show no visible tumor recurrence. Labels in the Pre-operative endoscopic images: d, nasal septum; e, soft palate; f, right torus tubarius; g, nasopharyngeal carcinoma; Post-operative: d, nasal septum; e, soft palate; f, nasopharyngeal right wall; g, clivus. (I) Pre-operative MRI shows the tumor invading the base of the skull (rT3); (J) post-operative MRI after six months did not show tumor recurrence; (K and L) The 6-months post-operative endoscopic examination images show no visible tumor recurrence. Labels in the pre-operative endoscopic images: h, nasal septum; i, nasopharyngeal posterior wall; j, right torus tubarius; k, nasopharyngeal carcinoma; Post-operative: h, nasal septum; i, nasopharyngeal posterior wall; j, nasopharyngeal right wall; k, clivus.

considered EBV-positive. EBV-DNA(+) refers to the pretreatment EBV-DNA-positive level unless otherwise stated.

## Follow-up and outcome measures

The patients began regular follow-up examinations after the treatment ended. During the first year after treatment, the patients returned to the hospital once every three months for examination as inpatients or outpatients. After 1 year, they received follow-up examinations once every six months, and after three years, they received follow-up examinations once

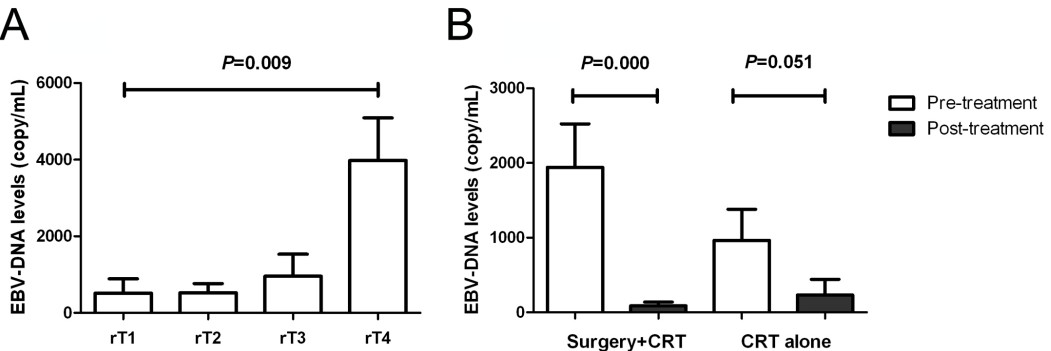

**Figure 2** **Correlation of EBV-DNA level with staging and treatment methods.** (A) Comparison of pre-treatment EBV-DNA levels among different T stages; (B) Changes in EBV-DNA levels between different treatment methods.

every 12 months. The date of the final follow-up visit was 30 July 2016; the median follow-up time was 31 months, and the follow-up rate was 100%.

## Statistical analysis

The SPSS 18.0 statistical software package (SPSS Inc., Chicago, IL, USA) was used to establish the database. Between-group comparisons of the count data were performed using the $\chi^2$ test. Pre- and post-treatment comparisons of EBV-DNA levels were performed using the Wilcoxon rank-sum test. Comparisons of EBV-DNA levels among different T-stages were performed using the Kruskal–Wallis $H$ test. Kaplan–Meier survival curves were used for univariate survival analysis, and the log-rank test was performed for the between-group comparison of survival curves. Multivariate analysis was performed using Cox regression analysis. $P < 0.05$ indicated that the differences were statistically significant.

## RESULTS

### Association of EBV-DNA with the clinical stages and changes in EBV-DNA pre- and post-treatment

The EBV-DNA positive rate in the serum of all patients was 43.55%. The mean EBV-DNA copy numbers of patients with stages rT1, rT2, rT3, and rT4 disease were 510, 583, 956, and 3,909 copies/mL, respectively. The EBV-DNA levels increased with NPC stages ($\chi^2$ value=11.674, $P = 0.009$) (Fig. 2A). The EBV-DNA levels decreased significantly in the surgery + post-operative CRT group ($Z = -3.484$, $P < 0.001$); compared with the pretreatment levels, the post-treatment EBV-DNA levels in the CRT group did not decrease significantly ($Z = -1.956$, $P = 0.051$) (Fig. 2B).

### Prognosis analysis

The 3-year overall survival (OS) for all patients was 51.40%, and the 3-year disease-free survival (DFS) was 46.86%. The univariate analysis indicated that local staging, treatment method, and pre- and post-treatment EBV-DNA levels correlated with the OS and DFS (all $P < 0.05$) (Table 2). Kaplan–Meier survival analysis indicated that the prognosis of patients with local early-stage NPC (T1 + T2) was significantly better than that of

**Table 2  Univariate analysis of impact on prognosis.**

| Factor | OS | | DFS | |
|---|---|---|---|---|
| | $\chi^2$ value | *P* value | $\chi^2$ value | *P* value |
| Sex | 0.235 | 0.627 | 0.935 | 0.333 |
| Age group | 0.618 | 0.432 | 1.576 | 0.209 |
| Recurrence time | 0.280 | 0.597 | 0.002 | 0.965 |
| Local staging | 8.954 | 0.030 | 8.194 | 0.042 |
| Treatment method | 4.054 | 0.044 | 7.019 | 0.008 |
| Pre-treatment EBV-DNA | 9.833 | 0.002 | 5.598 | 0.018 |
| Post-treatment EBV-DNA | 13.165 | <0.001 | 19.371 | <0.001 |

Notes.
Abbreviations:  OS, Overall survival; DFS, Disease-free survival.

**Table 3  Multivariate analysis of impact on prognosis.**

| Factor | OS | | DFS | |
|---|---|---|---|---|
| | HR | *P* value | HR | *P* value |
| Local staging | 1.515 | 0.043 | 1.685 | 0.004 |
| Treatment method | 0.468 | 0.043 | 0.393 | 0.008 |
| Pre-treatment EBV-DNA | 2.504 | 0.026 | NA | NA |

Notes.
Abbreviations:  OS, Overall survival; DFS, Disease-free survival..

patients with local late-stage NPC (T3 + T4) (Figs. 3A and 4A). The prognosis of the surgery + CRT group was superior to that of the CRT alone group (Figs. 3B and 4B). Pre- and post-treatment EBV-DNA levels in the peripheral blood were associated with the clinical prognosis (Figs. 3C, 3D, 4C and 4D). Notably, the patients with post-treatment EBV-positive disease were found to had obviously worse outcomes (Figs. 3D and 4D), which indicated that post-treatment EBV-positive strongly suggests a poor prognosis in patients with residual or recurrent NPC. Multivariate analysis indicated that local staging, pretreatment EBV-DNA levels, and the treatment method were independent risk factors for OS, whereas local staging and the treatment method were independent risk factors for DFS (Table 3). By the end of our follow-up, a total of 29 patients had died, of whom17 died of local recurrence, seven died of distant metastasis, three died of internal carotid artery rupture caused by skull necrosis, and two died of cervical recurrence.

## Subgroup analysis

The patients were divided into subgroups on the basis of local staging and the treatment method. The four subgroups were the local early-stage (T1 + T2) and surgery + CRT subgroup (17 cases), the local early-stage (T1 + T2) and CRT alone subgroup (12 cases), the local late-stage (T3 + T4) and surgery + CRT subgroup (19 cases), and the local late-stage (T3 + T4) and CRT alone subgroup (14 cases). The results indicated that for the local early-stage subgroups, the OS and DFS did not show significant differences among the different treatment methods (all $P > 0.05$) (Figs. 5A and 5B). For the local late-stage

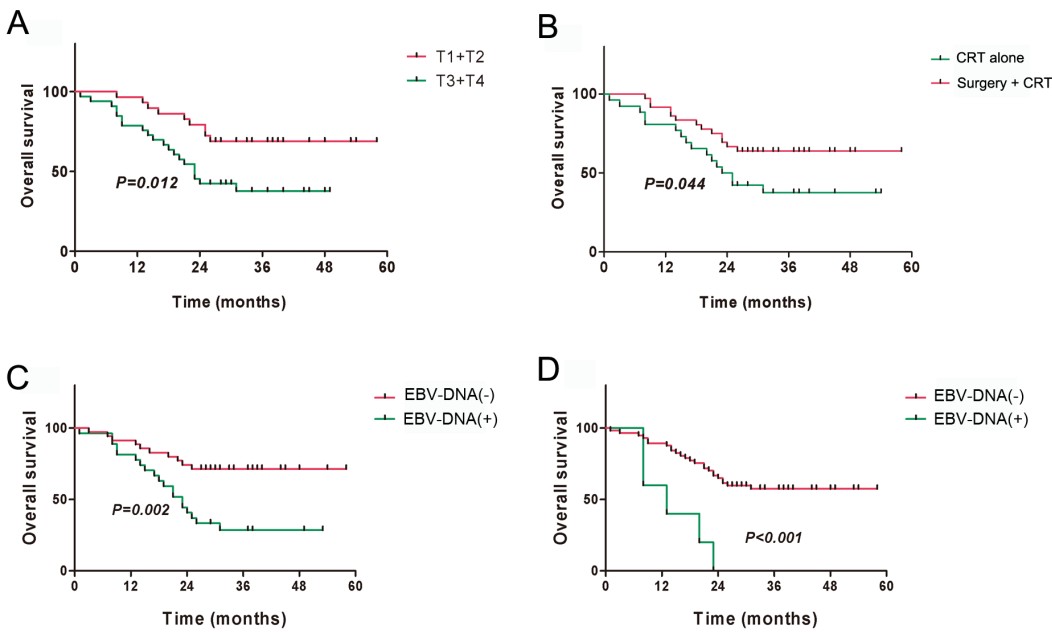

**Figure 3   Kaplan–Meier analysis of overall survival among patients with residual or recurrent NPC.**
(A) T stages; (B) Treatment methods; (C) Pre-treatment EBV-DNA; (D) Post-treatment EBV-DNA.

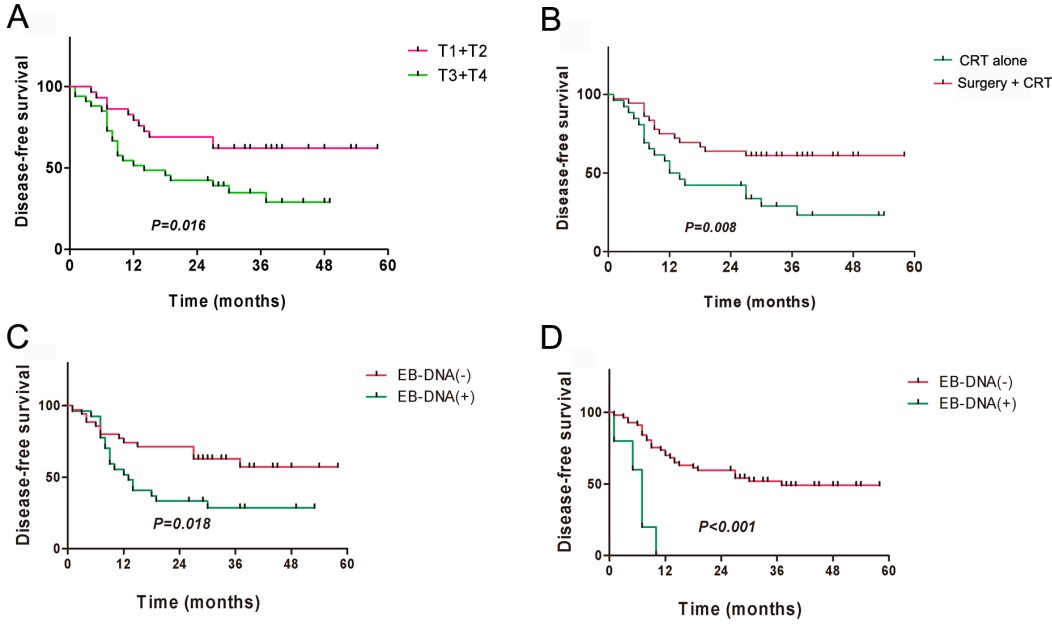

**Figure 4   Kaplan–Meier analysis of disease-free survival among patients with residual or recurrent nasopharyngeal carcinoma.** (A) T stages; (B) Treatment methods; (C) Pre-treatment EBV-DNA; (D) Post-treatment EBV-DNA.

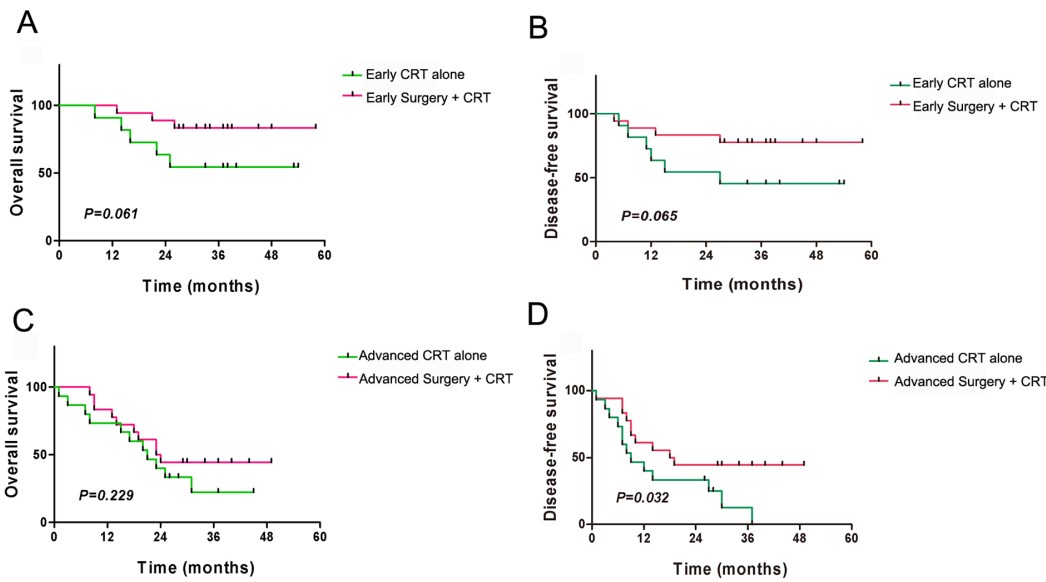

**Figure 5** **Kaplan–Meier analysis of clinical prognosis among subgroups of patients with residual or recurrent nasopharyngeal carcinoma.** (A) Overall survival (OS) comparison of different treatment methods in local early-stage patients; (B) Disease-free survival (DFS) comparison of different treatment methods in local early-stage patients; (C) OS comparison of different treatment methods in local late-stage patients; (D) DFS comparison of different treatment methods in local late-stage patients.

subgroups, surgery + CRT showed a better DFS than CRT alone ($P = 0.032$), whereas OS did not show a significant difference ($P > 0.05$) (Figs. 5C and 5D).

The patients were also divided into subgroups on the basis of the presence of EBV-DNA and the treatment method. The four subgroups were the EBV-DNA negative and surgery + CRT subgroup (19 cases), the EBV-DNA negative and CRT alone subgroup (16 cases), the EBV-DNA positive and surgery + CRT subgroup (17 cases), and the EBV-DNA positive and CRT alone subgroup (10 cases). The results indicated that the EBV-DNA negative and surgery + CRT subgroup had better OS and DFS than the CRT alone subgroup (OS: $P = 0.009$, DFS: $P = 0.003$) (Figs. 6A and 6B). Among the patients who tested positive for EBV-DNA, the OS and DFS did not show significant differences between the two treatment regimens (all $P > 0.05$) (Figs. 6C and 6D).

## Complications from surgery and chemoradiotherapy

In the surgical group, the most common complications from endoscopic surgery and/or radiotherapy were secretory otitis media (47.2%, 17/36), temporal lobe necrosis (25.0%, 9/36), skull base necrosis (19.4%, 7/36), cranial nerve palsy (13.9%, 5/36) and nasopharyngeal hemorrhage (5.6%, 2/36). In the non-surgical (CRT alone) group, the common complications from radiotherapy were temporal lobe necrosis (53.9%, 14/26), secretory otitis media (46.2%, 12/26), cranial nerve palsy (23.1%, 6/26), and skull base necrosis (15.4%, 4/26). The most common side-effects from chemotherapy in the both group were hematological toxicities, including leukopenia, neutropenia, and thrombocytopenia. Other non-hematological toxicities included mild transaminase

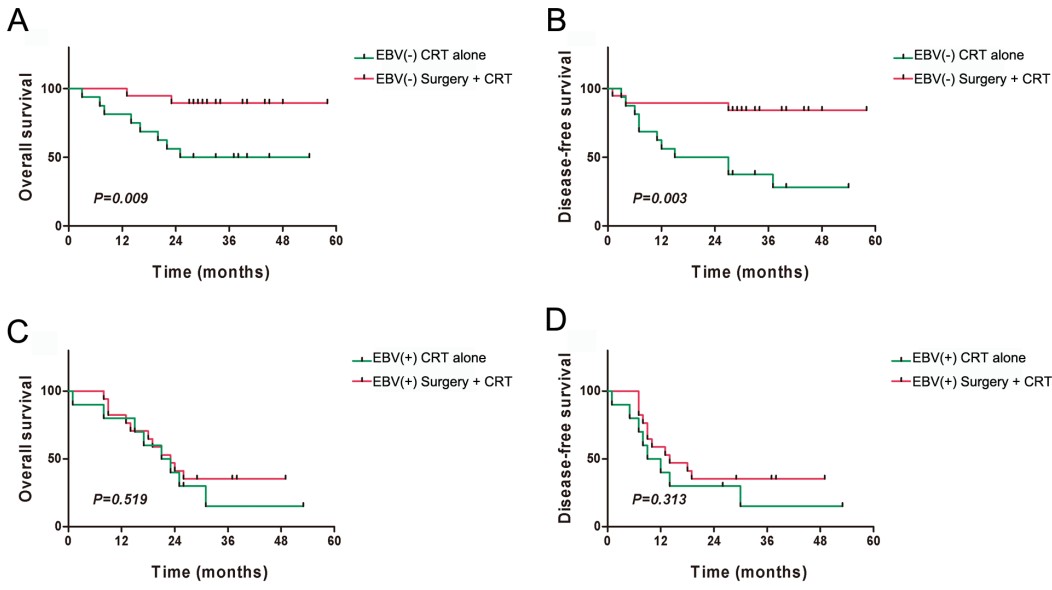

**Figure 6** **Kaplan–Meier analysis of clinical prognosis among the subgroups of patients with residual or recurrent nasopharyngeal carcinoma.** (A) Overall survival (OS) of different treatment methods among EBV-DNA(−) patients; (B) Disease-free survival (DFS) comparison of different treatment methods among EBV-DNA(−) patients; (C) OS comparison of different treatment methods among EBV-DNA(+) patients; (D) DFS comparison of different treatment methods among EBV-DNA (+) patients.

elevation and tolerable nausea and vomiting. In general, the chemotherapy-related side effects were tolerable and could be well-controlled.

## DISCUSSION

NPC is a common malignant tumor found in the Guangdong and Guangxi regions of China. According to the World Health Organization (WHO) pathological classification of tumors, NPC can be divided into WHO types I, II, and III. The most common pathological type is undifferentiated, non-keratinizing carcinoma (WHO type III), which is also associated with EBV infection. The EBV-DNA levels are related to the stage, volume, and prognosis of NPC at the initial treatment. Previous studies have shown that EBV-DNA levels are not detectable after treatment in patients with a first diagnosis of NPC, whereas detectable levels of EBV-DNA in the blood often indicate the presence of residual or recurrent NPC (*Lin et al., 2004*). In our patient group, the detection rate of EBV-DNA was 43.55%. With regard to the association between EBV-DNA levels and the stages of recurrent NPC, a study has found that EBV-DNA levels are related to tumor staging (*Chan & Wong, 2014*). Furthermore, on the basis of our data, the average EBV-DNA load for stage rT4 was 3,908 copies/mL, whereas that at stage rT1 was only 510 copies/mL. These results indicated that EBV-DNA may facilitate the assessment of clinical stages, in agreement with findings from a previous report (*Chang et al., 2012*). Further analysis revealed that the prognosis of patients who tested negative for EBV-DNA was superior to that of the EBV-DNA-positive group. Thus, when clinically evaluating a patient's condition, we recommend assessment of patient EBV-DNA levels in addition to clinical staging. A recent study has shown that EBV-DNA is

an independent risk factor for the prognosis of NPC at the time of initial treatment (*Peng et al., 2016*). The recent National Comprehensive Cancer Network(NCCN) guidelines have also recommended including EBV-DNA in the staging of NPC at the initial treatment. According to our present results, we agree that EBV-DNA should also be included in the staging of recurrent NPC.

A subsequent analysis has shown that patients with residual or recurrent NPC who test negative for EBV-DNA show better responses after surgery, whereas patients who test positive for EBV-DNA show similar responses after surgical treatment and CRT. However, patients receiving repeat radiotherapy have a poorer quality of life than those who undergo surgery (*You et al., 2015*). A number of studies have supported the claim that patients who test negative for EBV-DNA have better prognoses (*Shen et al., 2015*; *Stoker et al., 2016*), and have advocated surgery as the main treatment method (*You et al., 2015*). Unlike other studies in which the pretreatment EBV-DNA levels of recurrent NPC have been found to be predominantly positive (*Chan et al., 2012*), only 27 patients tested positive for EBV-DNA in our patient group. Therefore, the finding that the prognosis of the EBV-DNA-positive group was not better in the surgical group than in the non-surgical group may have been because of the small number of cases analyzed.

Unlike NPC at the initial diagnosis, the preferred treatment for recurrent NPC is often surgery, because it provides a better prognosis than CRT (*You et al., 2015*). Furthermore, in terms of resectable lesions, surgery is essentially able to achieve complete resection of the observable tumor. This study found that the EBV-DNA levels of the surgical group had decreased significantly, whereas those of the CRT alone group did not decrease significantly. Some studies have shown that blood EBV originates from tumor cells (*Lin et al., 2004*; *Wang et al., 2010*). It is known that plasma EBV levels are proportional to the viral production and viral release from the tumor cells and inversely proportional to the clearance rate of blood circulation. Therefore, the surgical resection of the tumor would lead to a significant reduction in EBV, thus causing a rapid decline in EBV-DNA levels. Through the continuous monitoring of post-operative EBV levels, researchers have found that the median clearance time is 139 min (*To et al., 2003*). Data from follow-up visits have revealed that patients with high pretreatment EBV-DNA levels have poorer prognoses (*An et al., 2011*). Furthermore, the continuous monitoring of EBV-DNA levels has shown that its clearance rate is related to the prognosis of recurrent NPC (*Wang et al., 2010*). In this study, EBV-DNA levels showed no significant differences between the surgery group and the CRT group before treatment; however, they decreased significantly after surgical treatment. This finding further suggests that surgical treatment was more effective than CRT at facilitating the removal of EBV, thus decreasing the probability of reactivating EBV. This EBV removal may be one of the reasons why surgical treatment had better therapeutic efficacy than CRT alone. However, surgery is not feasible in every case of recurrent nasopharyngeal carcinoma. For instance, some patients have contraindications for endoscopic nasopharyngectomy, such as internal carotid artery encasement, massive intracranial intradural involvement, and orbital content invasion (*Castelnuovo et al., 2013*). Thus, only the carefully chosen patients would benefit from surgery.

Staging is an independent risk factor for patients with recurrent NPC (*Hua et al., 2012*). We obtained similar results in our patient group, which showed that early-stage patients had better prognoses than late-stage patients. For early-stage patients, the surgical treatment had definite therapeutic efficacy, whereas CRT also achieved a satisfactory effect. For late-stage patients, the effective dose at the tumor center is relatively low, and radiotherapy often leads to residual tumor. Our data showed that among patients with local late-stage NPC, those who underwent surgical treatment had superior DFS, as compared with those who received CRT only ($P = 0.032$). This finding indicates that surgical treatment is still suitable for patients with local late-stage NPC and that it can achieve better therapeutic efficacy than non-surgical treatment. There were no significant differences in OS between the two groups, possibly because the patient deaths occurred because of other complications such as massive epitasis. Most previous studies have reported that surgery should be performed only in early-stage recurrent NPC. However, a recent study has shown that surgery has good efficacy for local intermediate-stage NPC (*Wong et al., 2017*).

Currently, the indications for surgical treatment and CRT in local recurrent NPC have not yet been clearly defined. In contrast, surgery is advocated as the main treatment method for the recurrence of lymph node metastases, because they can essentially be completely resected surgically and rarely leads to severe complications. However, complete surgical resection may not be achieved in NPC because of the complex anatomical location of the nasopharynx, the presence of important nerves and vessels in the vicinity, and the wide area of invasion in local late-stage recurrence. Therefore, previous studies have advocated that surgical treatment should mainly be reserved for rT1-T2 patients (*You et al., 2015*). With regard to local early-stage NPC, a previous study has reported that CRT can achieve satisfactory efficacy (*Qiu et al., 2012*). That study has also shown that the therapeutic efficacies of surgery and CRT are similar in early-stage patients. However, there is a lack of randomized control trials comparing surgical and non-surgical treatments and a paucity of relevant meta-analyses.

Our study was a single-center and small-cohort study, the findings presented in our study should be viewed as exploratory and needs to be further confirmed in subsequent studies. Our study was also a retrospective study in which the group allocation was based on the patient treatment wishes and with surgical contraindications or not, a design that may have led to bias. For ethical considerations, it may be difficult to perform a completely randomized controlled clinical trial to evaluate the additional benefits in the patients with residual or recurrent NPC undergoing endoscopic nasopharyngectomy combined with chemoradiotherapy compared with those receiving chemoradiotherapy alone. Even so, a prospective long-term follow-up and well-balanced cohort studies involving a larger sample number from multicenter should be conducted to more carefully evaluate the superiority of endoscopic surgery in addition to CRT in patients with residual or recurrent NPC.

## CONCLUSIONS

On the basis of the findings presented in our study, we suggest that serum EBV-DNA load is related to the stage of recurrent NPC. The combination therapy preceded by surgery can

effectively decrease the copy number of EBV-DNA, and its efficacy is superior to that of conventional CRT alone. Patients with local intermediate and late-stage NPC, especially those who test negative for EBV-DNA, may consider opting for surgical treatment followed by post-operative adjuvant radiotherapy or chemotherapy.

## ACKNOWLEDGEMENTS

We are thankful for the participation of every enrolled patient in this study.

### Funding

This work was supported by the Guangxi Natural Science Foundation (No. 2016GXNSFCB380003 and No. 2016GXNSFBA380144), Grant form Key Laboratory of Ministry of Education, Early Stage Prevention and Control of Regional high-incidence cancer in Guangxi (No. GJK201601), Scientific Research Project of Guangxi Health and Family Planning Commission (No. Z2013402), Guangxi Scientific Research and Technology Development Project (No. 14124003-1-3). There was no additional external funding received for this study. The funders had no role in study design, data collection and analysis, decision to publish, or preparation of the manuscript.

### Grant Disclosures

The following grant information was disclosed by the authors:
Guangxi Natural Science Foundation: 2016GXNSFCB380003, 2016GXNSFBA380144.
Key Laboratory of Ministry of Education, Early Stage Prevention and Control of Regional high-incidence cancer in Guangxi: GJK201601.
Guangxi Health and Family Planning Commission: Z2013402.
Guangxi Scientific Research and Technology Development Project: 14124003-1-3.

### Competing Interests

The authors declare there are no competing interests.

### Author Contributions

- Jingjin Weng and Jiazhang Wei conceived and designed the experiments, analyzed the data, wrote the paper, prepared figures and/or tables, reviewed drafts of the paper.
- Jinyuan Si, Yangda Qin, Min Li and Fei Liu analyzed the data, reviewed drafts of the paper.
- Yongfeng Si and Jiping Su conceived and designed the experiments, reviewed drafts of the paper.

### Human Ethics

The following information was supplied relating to ethical approvals (i.e., approving body and any reference numbers):

This retrospective study was approved by the ethics committee of the People's Hospital of Guangxi Zhuang Autonomous Region (Ethical Application Ref: Keyan-Guangxi-Keji-2016-31).

## Data Availability

The raw data is included as a Supplemental File.

## Supplemental Information

Supplemental information for this article can be found online at http://dx.doi.org/10.7717/peerj.3912#supplemental-information.

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
