# Peer review of "Clinical outcomes of residual or recurrent nasopharyngeal carcinoma treated with endoscopic nasopharyngectomy plus chemoradiotherapy or with chemoradiotherapy alone: a retrospective study"

_PeerJ, doi:10.7717/peerj.3912_

## Round 0.1 · original submission · Minor Revisions

· Academic Editor

Minor Revisions

Thank you for your submission. The reviewers and I feel that this article will be a valuable addition to the scientific literature with some modifications. See reviewer comments below. Several major themes in their comments that must be addressed:
1. The article should be reviewed by a native English speaker, as there are some grammatical errors and irregularities
2. The authors should include information about how treatment was selected for each patient - why did some patients have surgery and others did not?

Reviewer 1 ·

Basic reporting

1) The article should be reviewed by a native English speaker – there are significant grammatical errors.

Experimental design

1) Their inclusion criteria did not require a biopsy confirmation that the residual/recurrent mass was actually tumor. This seems a major limitation of the analysis. They also do not describe how many patients in the surgical group had false positive resections.
2) In the results section it is unclear whether the stage described refers to the original stage (as per standard practice) or the stage at recurrence. Based on Figure 2, it seems they are referring to recurrent Tstage. As a practice, these are referred to with Arabic numerals (1,2,3,4) , not roman numbers (I,II, III,IV).

Validity of the findings

3) The subgroup analysis involved very few patients – I am not sure how much we can rely on the validity of the data at that level. A comment should be made to indicate these findings should be viewed as exploratory. For instance, looking at the OS and DFS curves in Figure 5, the only group that reaches significance is surgery vs CRT for advanced disease.

4) The inclusion of EBV-DNA is fascinating and a clear strength of this paper. Given that the bloodtests were drawn serially, however, efforts should be made to clarify at what time point we are calling someone “EBV-positive”. In addition, more space should be used to emphasize the most impressive finding on these survival curves – the absolutely shockingly poor outcomes among patients with EBV + disease posttreatment. Again, though, does the EV+ in Figure 5 reflect EBV+ before or after treatment?

·

Basic reporting

The paper is well written and the conclusions reasonable.

Experimental design

The study design is effective and it is able to investigate the impact of EBV-DNA in patients' prognosis according to a subgroups stratification analyzing the stage of disease and the treatment strategy adopted.

Validity of the findings

Data is robust and statistically sound.

Additional comments

The paper can be improved by addressing these issues:
1) Please, specify in the material and methods section the histological subtype of nasopharyngeal cancer enrolled in this study. Only non-keratinizing carcinoma (WHO type III) have been included? I strongly suggest you to specify this information.
2) The results herein reported suggest that surgery was more effective than CRT at facilitating the reduction of EBV-DNA viral load and therefore surgical treatment has better therapeutic efficacy than CRT alone. (Discussion, lines 254-256). However, surgery is not feasible in every case of nasopharyngeal cancer recurrence (e.g. internal carotid artery encasement, massive intracranial
intradural involvement, orbital content invasion). I would like to suggest the Authors to improve the Discussion section by including the list of contraindications for endoscopic nasopharyngectomy currently accepted. In this regard, I suggest you to refer the study by Castelnuovo et al. where actual contraindications for salvage surgery in recurrent nasopharyngeal cancers have been clearly elucidated (Castelnuovo et al. Endoscopic endonasal nasopharyngectomy in selected cancers. Otolaryngol Head Neck Surg. 2013 Sep;149(3):424-30. doi:10.1177/0194599813493073)
3) A limit of this study is that patients have been allocated between the surgical and non-surgical group arbitrarily, without any randomization. The Authors should discuss deeply this and possibly propose future studies where a randomization can be introduced.

Reviewer 3 ·

Basic reporting

The article contains several English irregularities, especially with the use of adverbs ( see examples in the revised PDF). It would benefit from a revision by a native English speaker.
Overall, it is well organized and beautifully illustrated

Experimental design

The main flaw in this section is the lack of clarity about how the treatments were chosen. It lacks a description of criteria used in each group. There are other important data points missing (see revised PDF)
Include reason for death (local, loco regional, distant mets, side effects of treatment.

Validity of the findings

The analysis seems sound; however, the conclusions are flawed in view of the lax methodology, as described above. They will have to be revised.
In addition, theaters fail to address complications from each approach of treatment. This can also affect

Annotated reviews are not available for download in order to protect the identity of reviewers who chose to remain anonymous.

---

## Round 0.2 · accepted · Accept

· Academic Editor

Accept

Thank you for addressing the concerns of our reviewers.